# Simulating the progression of the COVID-19 disease in Cameroon using SIR models

Ulrich Nguemdjo[1,2]*, Freeman Meno[3], Audric Dongfack[4], Bruno Ventelou[1]

1 AMSE, Centrale Marseille, EHESS, CNRS, Aix-Marseille University, Marseille, France, 2 Laboratoire Population—Environnement—Développement, Aix-Marseille University, Marseille, France, 3 Lycée Polyvalent Franklin Roosevelt, Reims, France, 4 Ecole Centrale Marseille, Aix-Marseille University, Marseille, France

* ulrich-boris.nguemdjo-kamguem@univ-amu.fr

## Abstract

This paper analyses the evolution of COVID-19 in Cameroon over the period March 6–April 2020 using SIR models. Specifically, we 1) evaluate the basic reproduction number of the virus, 2) determine the peak of the infection and the spread-out period of the disease, and 3) simulate the interventions of public health authorities. Data used in this study is obtained from the Cameroonian Public Health Ministry. The results suggest that over the identified period, the reproduction number of COVID-19 in Cameroon is about 1.5, and the peak of the infection should have occurred at the end of May 2020 with about 7.7% of the population infected. Furthermore, the implementation of efficient public health policies could help flatten the epidemic curve.

## 1. Introduction

The new coronavirus (COVID-19) started in Wuhan, China last November, presenting pneumonia-like symptoms in patients. The first cases examined in China indicated that it was a new respiratory disease. However, The World Health Organization (WHO) began delivering the most recent discoveries related to this virus on January 7, 2020. By the end of January 2020, with the virus already spread to several countries, the WHO alerted the world, announcing that COVID-19 was an international sanitary crisis.

On February 14, 2020, the WHO [1] reported the first confirmed case on the African continent in Egypt, and the Ministry of Public Health of Cameroon announced their first confirmed case on March 6, 2020 [2]. Following this announcement, the spread of the disease gradually increased within the population of Cameroon. Based on the current situation, there is a need to study the evolution of this virus and the efficiency of the measures adopted by local authorities to curb the spread of COVID-19 in Cameroon. The core of the article will give details on the successive steps that we followed for the forecasting exercise: collecting data from the Cameroonian Ministry of Public Health, designing and applying the model, performing simulations for varying rates of exposure and transmittance, and evaluating the results obtained.

**Data Availability Statement:** The proposed research includes daily data on the number of confirmed cases, the number of deaths, and the number of recovered from the COVID-19 pandemic going from March 6th, 2020 to April 10th, 2020.

Our data have been collected from daily tweets of the public health minister, the daily report on the website of the ministry, and daily Facebook/twitter post of the main Cameroonian National media (Cameroon Radio Television, CRTV). As the data are collected from public access, we do not have any restrictions on making it available. The following link gives access to our dataset: https://doi.org/10.6084/m9.figshare.12613649.v1

**Funding:** The authors received no specific funding for this work.

**Competing interests:** The authors have declared that no competing interests exist

## 2. Data

To explore the evolution of the coronavirus disease in Cameroon, the present paper uses data collected from the Cameroonian Ministry of Public Health. According to the government, the first confirmed case of COVID-19 was detected on March 6, 2020, in the capital, Yaoundé. The government decided to publish daily reports regarding the evolution of the disease including the number of confirmed cases of COVID-19, the number of deaths due to the virus, and the number of recoveries via public declaration. These daily declarations ended on April 10, 2020, which is why our dataset covers from March 6 to April 10.

Fig 1 displays an overview of the evolution of the coronavirus disease in Cameroon during our period of study. It shows an exponential increase in the number of confirmed cases. The total number of confirmed cases increases from 10 on March 17 to 820 on April 10, multiplying the total number of cases by 82 in less than a month. Moreover, the figure shows a slight increase in the total number of deaths due to COVID-19 in the country, from 1 on March 25 (the first death) to 12 on April 10. On the last day of our period of observation, the figure shows that the total number of recoveries from COVID-19 in Cameroon is 98, around 8 times more than the total number of deaths observed on the same date.

## 3. The SIR model

To explore the evolution of the coronavirus disease in Cameroon, this paper uses a simple SIR (susceptible (S)–infectious (I)–recovered (R)) model known as a general stochastic epidemic model [3–6]. This model is particularly suitable when dealing with a large population [7]. The initial model considers that individuals are at first Susceptible. If they are infected by the virus, they become immediately Infectious, and they remain infectious until they Recover, assuming immunity during the rest of the outbreak. In our paper, the last group was modified to "Removed" in order to account for the individuals who can no longer transmit the disease for reasons other than recovery, such as placement in quarantine, hospitalization, or dying from the virus [8]. To date (April 10, 2020), it is not certain whether, in the case of COVID-19, the recovered will be immunized for a very long period. In this paper, we assume that recovery confers immunity during the rest of the outbreak. We also assume that the population in our study is closed and that individuals mix uniformly in the community. Following Britton and Giardina [9], we also assume that all individuals are equally susceptible to the disease and equally infectious if they get infected.

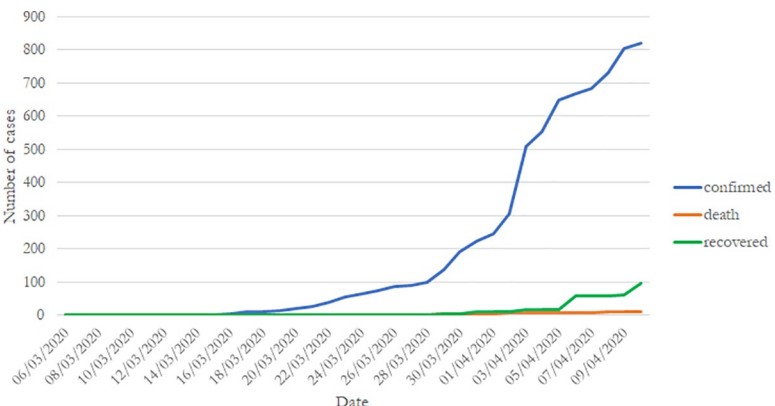

**Fig 1. Number of observed cases of COVID-19 in Cameroon.**

Supposing a closed population of size N, an individual who gets infected by the coronavirus becomes immediately infectious and remains so for a time determined exponentially with a decay rate of $\gamma$ (the removal rate). Thus, $\gamma^{-1}$ refers to the average number of infectious days that an infected individual has to transmit the virus before being removed from the Infectious group (placed in quarantine, hospitalized, recovered, or died). During the infectious period, an individual has close contact with the rest of the population over the remaining time at a rate determined by the parameter $\beta$ (also known as effective contact rate), which is the product of the average number of exposures per unit of time ($\tau$) and the likelihood of infection at each occasion of exposure ($\mu$) [3].

$$\beta = \tau \times \mu \tag{1}$$

$S(t)$, $I(t)$, and $R(t)$ respectively represent the number of susceptible, infectious, and removed individuals in the population at time $t$. Assuming that we have a closed population, the total population at time $t$ is given as follows:

$$N(t) = S(t) + I(t) + R(t) \tag{2}$$

Let's assume that at the beginning of the epidemic $(S(0), I(0), R(0)) = (N-1, 1, 0)$ meaning that there is initially one infectious individual in the population and no removal. All other things being equal, the number of susceptible individuals decreases symmetrically by:

$$\frac{dS}{dt} = -\beta \frac{S(t)I(t)}{N} \tag{3}$$

Also, the variation in the number of people infected according to this measure is given by:

$$\frac{dI}{dt} = \beta \frac{S(t)I(t)}{N} - \gamma I(t) \tag{4}$$

The result is the following dynamic system, reflecting the generalized SIR Model:

$$\begin{cases} \dfrac{dS}{dt} = -\beta \dfrac{S(t) \times I(t)}{N} \\ \dfrac{dI}{dt} = \beta \dfrac{S(t) \times I(t)}{N} - \gamma I(t) \\ \dfrac{dR}{dt} = \gamma I(t) \end{cases} \tag{5}$$

The simplicity of this dynamic system gives us rapid information on the rate of spread of the epidemic. Indeed, an epidemic occurs if the number of infected individuals increases continuously, i.e., $\frac{dI}{dt} > 0$

$$\frac{\beta}{\gamma} \times \frac{S(t)}{N} > 1 \tag{6}$$

As highlighted by Jones [3], at the outset of an epidemic, nearly everyone is susceptive. Thus, $\frac{S(t)}{N}$ can be approximated to 1 and the above equation can be written as:

$$\frac{\beta}{\gamma} = R_0 > 1 \tag{7}$$

### 3.1. The parameter $R_0$

The parameter $R_0$ is called the basic reproduction rate. It is the expected number of secondary cases produced by a single infectious individual during the infection period in a completely susceptible population [3, 10]. Regarding the value of the parameter, the severity of the epidemic can be summarized into two cases [11, 9]:

- $R_0 > 1$: The Supercritical case. The epidemic increases exponentially: one infected individual infects more than one individual on average.

- $R_0 \leq 1$: Critical case. No epidemic: the disease will surely die out without affecting a large share of the population

  Note that an $R_0 > 1$ does not always guarantee an epidemic in the population [12, 9].

## 4. The results

### 4.1. Estimating the parameters of the model

To simulate the progression of the coronavirus disease in Cameroon, the next step consists in determining the parameters $\beta$ and $\gamma$ that best describe the current evolution of the virus in the country presented in section 2. For this step, knowing the period of observation, we chose to neglect the slight measures adopted by the Cameroonian local government to restrain the spread of the virus.

The estimation is done using a Nelder-Mead and maximum likelihood optimization algorithm. The process implies finding $\beta$ and $\gamma$ in such a way that when these parameters are substituted in the equations described above, the difference between the data obtained and that recorded on the field is minimized. Computing $\beta$ and $\gamma$ using data recorded from March 6 to April 10 and a population of size N = 25,216,237. The results are summarized in Table 1. The confidence intervals presented in the table are built using the distribution of the bootstrap realizations presented in the Appendix (Fig 2, Fig 3, Fig 4, Fig 5 and Fig 6). Simulations conducted with the above parameters yield the results shown in Fig 7.

All else unchanged, the evolution presented in Fig 7 should be a plausible scenario if no action is taken to reduce the spread of the virus. More precisely, the figure reveals that, if the situation remains as during the observed period (from March 6 to April 10), about 7.7% of the Cameroonian population might have ended up being infected, which is close to *2,015,200* individuals. In this case, the peak of the infection will occur between day number 78 and day number 85 starting from March 6, which is between 23th and 29th May. Additionally, assuming a rate of 15.8% of serious complications of the disease [13] approximately 318,402 Cameroonians might find themselves hospitalized in critical conditions. While considering a mortality rate of 3.4% (the overall mortality rate of COVID-19 announced by the WHO on March 3,

**Table 1. Estimates parameters.**

|  | Original | Bias | Std. error | Bootstrap normal CI[*] | |
|---|---|---|---|---|---|
|  |  |  |  | Inf | Sup |
| $\beta$ | 0.615 | 7.65e-06 | 0.003 | 0.610 | 0.619 |
| $\gamma$ | 0.393 | -3.69e-05 | 0.003 | 0.388 | 0.398 |
| $R_0$ | 1.567 | 0.000 | 0.016 | 1.536 | 1.597 |
| Maximum Infected | 2,015,200 | 757.6864 | 76,463.73 | 1,864,576 | 2,164,309 |
| Number of Days to reach the peak | 81,06 | -0.022 | 1.660 | 77.81 | 84.32 |

[*]CI = Confidence Interval.

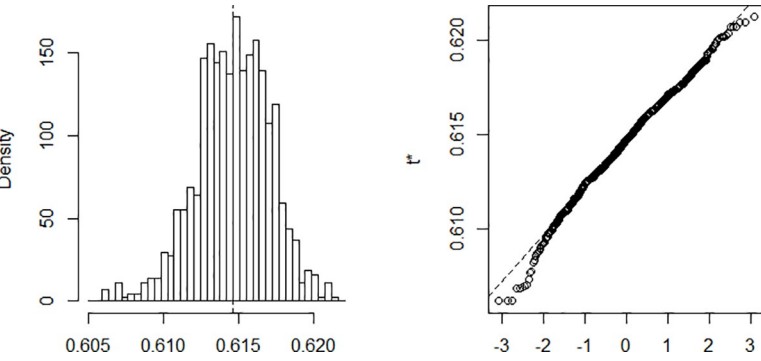

**Fig 2. Bootstrap distribution of β.**

2020 [14], the expected number of deaths due to coronavirus could be close to 68,517. Though these results might not be accurate, the approach gives a simple and quick means to figure out the evolution of the spread of the virus and paves the way for a better approach. Our model also suggests that the basic reproduction rate ($R_0$) of COVID-19 in Cameroon up to April 10 is about 1.567 persons, meaning that on average an infectious individual infects 1.567 susceptible individuals during his infection period.

The question now is how can we flatten the infectious curve? What has been recommended since the beginning of this epidemic is to apply public health measures.

## 4.2. Flattening the epidemic curve with public health interventions

Until an efficient medical treatment or vaccine for COVID-19 is available, prevention and control strategies to reduce or stop the transmission of the disease only rely on measures adopted by public health officials. In this section, we model the effect of different public health interventions on the spread of the coronavirus disease in Cameroon.

**4.2.1 Physical distancing.**   Physical or social distancing–keeping space between yourself and other people outside your home–plays a major role in public health interventions. Its objective is to reduce the probability of contact between infected individuals and susceptible ones to minimize the transmission of the disease. In practice, physical distancing [15] can be implemented by adopting the following habits:

- Stay at least 1 meter (3 feet) from others

- Do not gather in groups

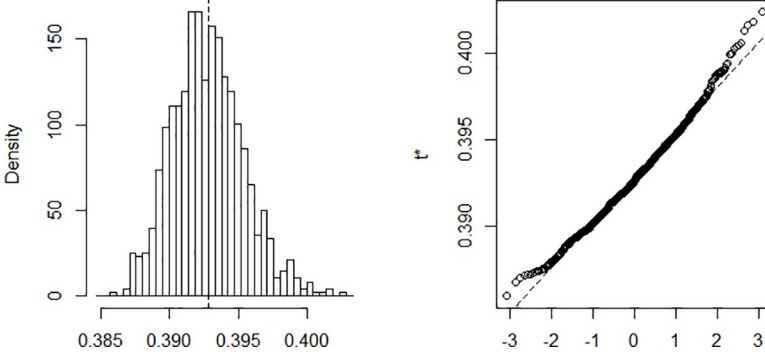

**Fig 3. Bootstrap distribution of γ.**

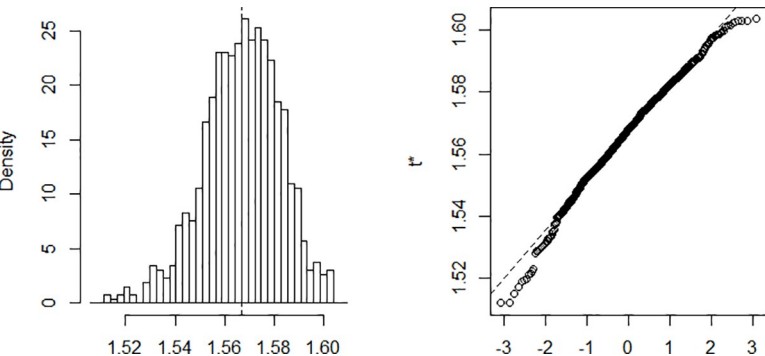

**Fig 4. Bootstrap distribution of $R_0$.**

- Stay out of crowded places

- Avoid mass gatherings

**4.2.2. Hygiene measures.** In addition to physical distancing, public health interventions also incorporate hygiene measures to reduce transmission of the coronavirus between individuals, from individuals to surfaces, and from surfaces to other individuals. These hygiene measures include:

- Frequent hand washing

- Avoiding touching eyes, nose, and mouth

- Practicing respiratory hygiene: cover the mouth and the nose with the bent elbow or tissue when coughing or sneezing

- Wearing surgical masks

- Cleaning or disinfection of fomites

**4.2.3. Simulating public health interventions.** The main concern of public health interventions is to flatten the infectious curve. The shape of the infectious curve is a function of the effective contact rate β, which depends on the average number of exposures per unit of time (τ) and the likelihood of infection at each occasion of exposure (μ) (see section 3). Thus, adjusting τ and μ will influence the kinetics of the flattening curve [16]. More precisely, to

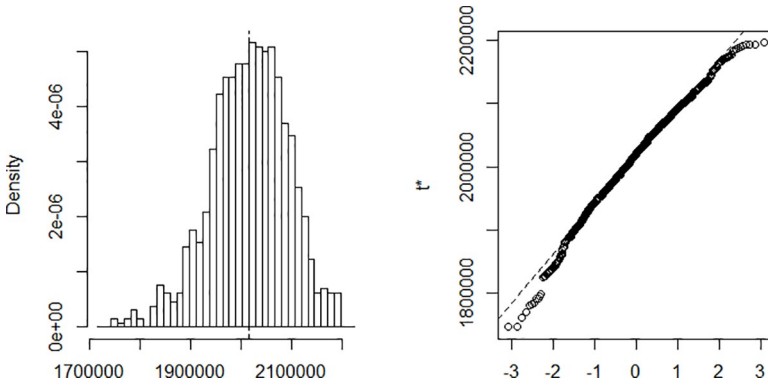

**Fig 5. Bootstrap distribution of the infected.**

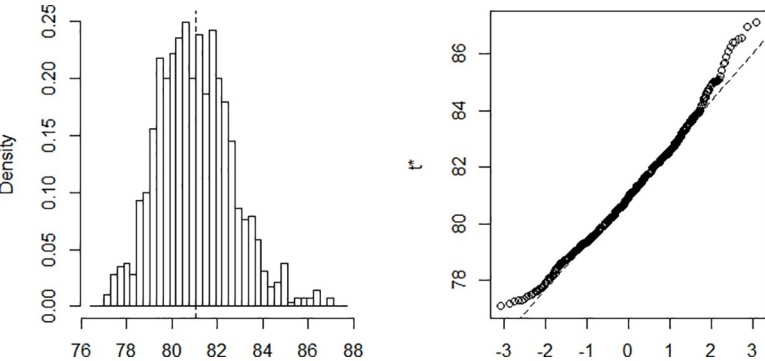

**Fig 6. Bootstrap distribution of the number of days.**

simulate an increase in social or physical distancing, we can reduce the average number of exposures per day ($\tau$), whereas, taking into consideration the hygiene measures, we can reduce the probability of infection at each occasion of exposure ($\mu$).

According to some studies on COVID-19, the probability of infection at each exposure varies from 1% to 5% [17]. In this paper, 5% is selected as the initial infection probability. As our parameter $\beta = 0.6$, we can estimate the initial average number of exposures per day to 12 exposures. Therefore, the goal of the public health interventions in Cameroon will be to progressively reduce the number of exposures (we will simulate using 12, 6, and 2 exposures per day) and the probability of infection (we will simulate using 5%, 2.5%, and 1%). The simulation was done using the R *EpiModel* package with a population of size N = 1,000 for computational matters [18]. Fig 8 gives the results.

As we can see, a decrease in the number of exposures per day (from the left to the right on Fig 8) and a decrease in the likelihood of infection (from the top to the bottom of Fig 8) is associated with a flattening in the coronavirus epidemic curve. Furthermore, as the curve flattens, we can observe a decrease in the percentage of the population infected by the virus. Also, decreasing exposures per day or the infection probability prolongs the epidemic overall while slowing down the incidence rate.

## 5. Conclusion

The present paper aims to analyze the evolution of the coronavirus disease in Cameroon. The study uses data collected by the Cameroonian health ministry between March 6 (date of the first confirmed case in the country) to April 10, 2020.

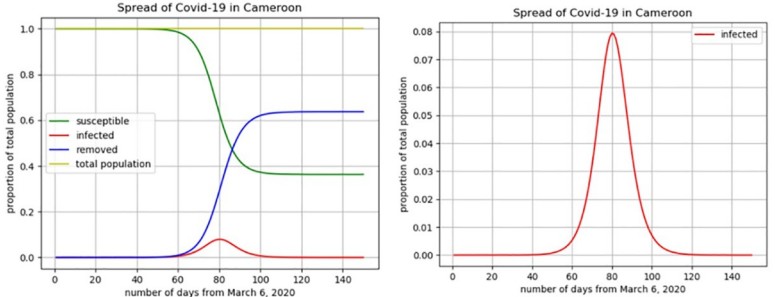

**Fig 7. Evolution of the coronavirus disease in Cameroon.**

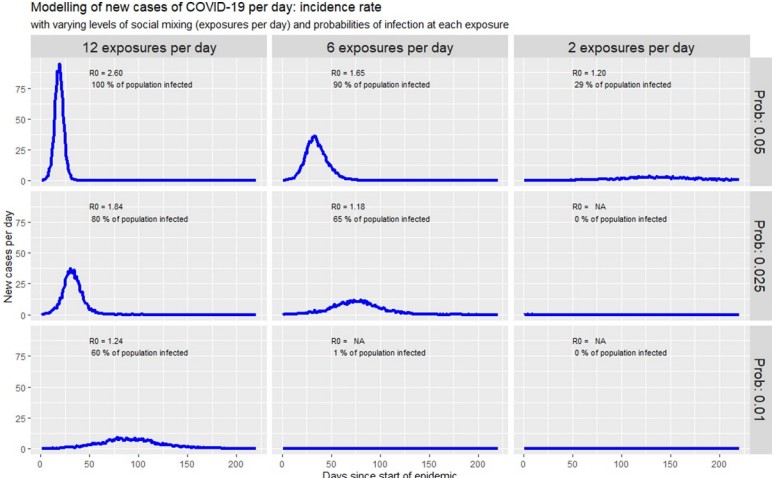

**Fig 8. Simulating new cases of COVID-19 with public health interventions.**

Descriptive statistics show an exponential increase in the total number of infectious individuals. This gives a first idea of how the virus is spreading out in Cameroon. Starting from the natural evolution of the epidemic in the first days of propagation in the country, a SIR model was applied to the observed data. The results of the calibration show that, if actions undertaken by Cameroonians to fight against the coronavirus do not improve, the peak of the infection would occur at the end of May 2020, with about 7.7% of the Cameroonian population infected. Using the WHO mortality rate associated with COVID-19, the expected number of deaths due to the virus in Cameroon could be close to 70,000. At this time, and using the most recent information available for Cameroon to verify the accuracy of the modeling, one can observe that the epidemic indeed reached its peak at the end of May 2020, as suggested by the public health minister in an alarmist tweet on May 25[th] [19]. For the first time on Twitter, the minister warned that the country was entering a complicated phase of the epidemic, and encourage preventive measures such as handwashing and wearing mask when going out. In later comments related to this tweet, he stated the peak was soon approaching [20].

However, by intensifying public health interventions, the epidemic curve could have flattened more, as suggested in the simulation. The results of the modeling seem to underline the value of appropriate communication campaigns from the government and the importance of the population's compliance with the public health measures recommended by the WHO to limit and stop the spread of the coronavirus disease, at least while waiting for possible preventive and/or curative treatments to be found. To date, some public measures have been taken by the Cameroonian government such as sensibilization through public media, shutting down schools, and the closing of public spaces and of some informal/formal businesses. However, these measures were relaxed just a few weeks after their execution due to their negative economic impact on most Cameroonian households. This quick renouncement to the measures may explain why the alarmist tweet of the health minister occurred exactly when the model predicted the peak without any intervention, precisely at the end of May [19].

Even though the paper has some limitations, such as the 'closed population' assumptions or the homogeneous mixture of the population (particularly across the geography of Cameroon), our model is particularly suitable while dealing with a large population of subjects, such as a human population observed at a country level [3]. We should also note that the basic reproduction rate, which is constant in this paper, may change depending on several variables

(awareness of the population, intensity of trade, movement of people), and thus, create a time-varying basic reproduction rate [21, 22]. In all cases, knowing the paucity of the literature available for African countries, this paper enriches the knowledge by providing some quantitative evidence in support of the Cameroonian government's actions attempting to fight against the coronavirus. As an extension of this paper, studies with more sophisticated assumptions on the contact-matrix could be carried out (the matrix that accounts for the different interactions between the susceptible), as well as simulations showing, in parallel of the disease propagation, the economic impact of the coronavirus epidemic on the Cameroonian population.

## Author Contributions

**Conceptualization:** Ulrich Nguemdjo.

**Data curation:** Ulrich Nguemdjo, Audric Dongfack.

**Methodology:** Ulrich Nguemdjo.

**Project administration:** Ulrich Nguemdjo.

**Software:** Freeman Meno, Audric Dongfack.

**Supervision:** Bruno Ventelou.

**Validation:** Ulrich Nguemdjo.

**Writing – original draft:** Ulrich Nguemdjo, Freeman Meno, Audric Dongfack.

**Writing – review & editing:** Bruno Ventelou.

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
