## [Decision Letter · Decision Letter 0]

16 Jun 2020

PONE-D-20-14463

Simulating the progression of the COVID-19 disease in Cameroon using SIR models

PLOS ONE

Dear Dr. NGUEMDJO,

Thank you for submitting your manuscript to PLOS ONE. After careful consideration, we feel that it has merit but does not fully meet PLOS ONE’s publication criteria as it currently stands. Therefore, we invite you to submit a revised version of the manuscript that addresses the points raised during the review process.

Your manuscript was reviewed by 2 experts in the field. Both identified many important problems in your submission, which require your careful attention. Please review the attached comments and provide point-by-point responses.

We look forward to receiving your revised manuscript.

Kind regards,

Yury E Khudyakov, PhD

Academic Editor

PLOS ONE

Journal Requirements:

4. Please ensure that you refer to Figures 4 to 9 in your text as, if accepted, production will need this reference to link the reader to the figure.

Additional Editor Comments (if provided):

Reviewers' comments:

Reviewer's Responses to Questions

**Comments to the Author**

1. Is the manuscript technically sound, and do the data support the conclusions?

Reviewer #1: Yes

Reviewer #2: Yes

2. Has the statistical analysis been performed appropriately and rigorously? 

Reviewer #1: Yes

Reviewer #2: Yes

3. Have the authors made all data underlying the findings in their manuscript fully available?

Reviewer #1: Yes

Reviewer #2: No

4. Is the manuscript presented in an intelligible fashion and written in standard English?

Reviewer #1: Yes

Reviewer #2: Yes

5. Review Comments to the Author

Reviewer #1: PLOS ONE

Manuscript number: PONE-D-20-14463

Title: Simulating the progression of the COVID-19 disease in Cameroon using SIR models

Nguemdjo et al. apply a modified “Susceptible-Infected-Recovered” epidemiological model to COVID-19 data collected from the Cameroonian Ministry of Health. They generate models with various degrees of infectiousness and infer upon to which level the implemented measures of public heatlh intervention would successfully contain the epidemic.

General comments:

The paper is confirming the validity of a globally applied epidemic model within the settings of Cameroon and providing valuable statistical information regarding the resolution of the pandemic crisis at a national level.

The authors may also wish to consider the following:

1. A general revision of English language, syntax and grammar is necessary.

2. Footnotes must instead be incorporated into the text or properly added as references following the journal’s citation style.

3. Figures should have captions in a style as dictated by the journal, with a more extensive legend.

4. Gallicisms, such as using the French spelling of the name of the Republic of Cameroon, should be corrected.

5. The Authors analyse data available until April 10th, and its association with public health interventions. Given that more than one month has passed after this date, the Authors can comment and provide more information on the current situation and the impact of the measures taken.

Specific comments:

1. “The rest of the study is organized…” - this details the structure of the paper (which is unnecessary), not the study itself. That would be collecting data from the Cameroonian ministry of health, applying the model, performing simulations for varying rates of exposure and transmittance and evaluating the data.

2. “The political capital, Yaoundé”

3. Line 41/54: The headings “The Model” and “The SIR Model” are redundant with each other. The sections should be merged.

4. Line 45: Provide a source for the sentence “This model..... a large population”.

5. Lines 45-46: When first explaining the model, names of the population fractions shouldn’t be capitalized (“Susceptible”, “Infectious”, “Recovered”), instead the abbreviation should be put in brackets: “susceptible (S)”, “infectious (I)”, “recovered (R)”.

6. Lines 47-48: “the last group was modified to “Removed”” - here a clearer explanation is needed, for example “was modified to “Removed” in order to cover/account for individuals who can no longer transmit the disease for reasons other than acquired immunity, such as being placed in quarantine, hospitalized, or dying”.

7. Line 56: “an exponentially distributed time” - Time is not distributed, but determined exponentially with a decay rate of γ.

8. Line 60: Why does ”close contact” need to be in quotes? Also, it should be merged with the next sentence: “… an individual has close contact with the rest of the population over the remaining time at a rate determined by the parameter β (also known as effective contact rate), which is the product of the average number of exposures per unit of time (τ) and the likelihood of infection at each occasion of exposure (μ).” The equation (β = τ × μ) should be included in its own paragraph below.

9. Line 107: Consider substituting “if nothing is done indeed” by “if no action is taken”.

10. Line 150-155: There lacks a clear, mathematical expression of the relationship between hygienic measures and the probability of infection upon contact. It needs to be framed in terms of the removal rate and its connection to flattened curve kinetics.

11. “Besides, we can observe a decrease in the percentage of the population infected by the virus” A direct link between a flatter curve, a greater proportion of Removed individuals and total population infected is not sufficiently made clear.

12. “and the disease lasts for a shorter period” - from my understanding decreasing exposures per day prolongs the epidemic overall while slowing down the incidence rate.

13. “a SIR model was calibrated” : “calibration” is usually understood as the preparation of a model or device for accurate measurements by adjusting it to known standards. In this case, “applied”, “simulated” or “fitted” would be suitable to use.

14. Line 190: “contact-matrix (between susceptible)” – upon introducing a new term, provide more detailed information about it.

15. Footnote number 10 should be incorporated in the methods section.

Minor spelling, grammar and punctuation issues:

1. Line 9 : “could help flattens the epidemic curve”, remove the “s” in flattens.

2. Line 16: The WHO alerted, instead of “alleged”.

3. Line 21 : “there is a need to study”

4. Line 22: “Covid-19 diseases” is redundant, remove the word diseases.

5. Line 29 : “The government decided to publish daily reports”

6. Line 30 : “the number of deaths”

7. Line 31 : “These declarations ended on April 10, 2020, that which is why…”. Remove “that”.

8. Line 44 : Remove “while” in “suitable while when dealing”

9. Line 48: “Thus, the individual is Infectious until he is” - “Thus, the individual is infectious until they are…”; Also line 121.

10. Line 55 : “Supposed” instead of “Supposing a closed population…”

11. Line 59 : Substitute word infectious by infection in “During the infectious period ”; Also line 80, 122.

12. Line 60 : “at a rate β”

13. Line 99 : “Marche 6”. Remove “e”

14. Line 157: “to progressively reduce instead of to reduce progressively”

15. Figure 1: “recovery” should be replaced by “recovered” or “recoveries”.

16. Figure 2: “total population”

17. Appendix: The “the” before letters denoting parameters should be removed (“distribution of the β”).

Reviewer #2: Summary: This is an interesting article on how to use SIR models to make forward decisions on COVID-19. The authors use data from Cameroon to fit the model parameters and then make predictions on how different control strategies will impact the dynamics.

Recommendations:

1. The citations are low and there are nearly daily publications on COVID-19 SIR models. Two are in NEJM and JAMA. The authors should cite these to let readers know this is standard.

2. The SIR model is the Kermack and McKendrick model not Kendrick “model known as a general stochastic epidemic model (Kendrick and Kermarck 1927; Jones 2007)”.

3. Make a table of parameter meanings, description of how they are estimated and their units. The parameters all have standard names like effective contact rate etc. The authors should rely on these standards.

4. The model doesn’t include any time lag and relies on those who were tested. Were these symptomatic cases? If so, the authors should point out their model does not include asymptomatic cases that were infectious and thus represents an underestimate.

5. The calculation of R0 can be done in different methods. The authors make an assumption which helps their calculation. They should cite Pauline van den Dreissche’s work on calculating R0. Also, R0 may change based on controls and thus create a time-varying R0. This should be noted in the discussion. A constant R0 is based on the assumptions behind the model. If these assumptions change, then the R0 changes.

6. There is no data sharing plan or information.

7. There are some places the language can be tightened up. The authors should review a few times to clean up the writing.

6. PLOS authors have the option to publish the peer review history of their article (what does this mean?). If published, this will include your full peer review and any attached files.

Reviewer #1: No

Reviewer #2: Yes: Diana Thomas

---

## [Author Response · Author response to Decision Letter 0]

6 Jul 2020

Dear Editor,

Thank you for giving us the opportunity to revise our paper on “Simulating the progression of the COVID-19 disease in Cameroon using SIR models” The suggestions offered by the reviewers have been helpful and we also appreciate your insightful comments on revising some aspects of the paper.

We have included the reviewers’ comments immediately after this letter and responded to them individually, indicating how we addressed each concern and describing the changes we have made.

Best regards,

Ulrich NGUEMDJO

Freeman MENO

Audric DONGFACK

Bruno VENTELOU

---

## [Decision Letter · Decision Letter 1]

23 Jul 2020

PONE-D-20-14463R1

Simulating the progression of the COVID-19 disease in Cameroon using SIR models

PLOS ONE

Dear Dr. NGUEMDJO,

Thank you for submitting your manuscript to PLOS ONE. After careful consideration, we feel that it has merit but does not fully meet PLOS ONE’s publication criteria as it currently stands. Therefore, we invite you to submit a revised version of the manuscript that addresses the points raised during the review process.

Your manuscript was reviewed by one of the original reviewers. There are  still problems that require your attention. Please note comments on careful spell checking your manuscript and using references.

We look forward to receiving your revised manuscript.

Kind regards,

Yury E Khudyakov, PhD

Academic Editor

PLOS ONE

Reviewers' comments:

Reviewer's Responses to Questions

**Comments to the Author**

1. If the authors have adequately addressed your comments raised in a previous round of review and you feel that this manuscript is now acceptable for publication, you may indicate that here to bypass the “Comments to the Author” section, enter your conflict of interest statement in the “Confidential to Editor” section, and submit your "Accept" recommendation.

Reviewer #2: (No Response)

2. Is the manuscript technically sound, and do the data support the conclusions?

Reviewer #2: Yes

3. Has the statistical analysis been performed appropriately and rigorously? 

Reviewer #2: Yes

4. Have the authors made all data underlying the findings in their manuscript fully available?

Reviewer #2: Yes

5. Is the manuscript presented in an intelligible fashion and written in standard English?

Reviewer #2: No

6. Review Comments to the Author

Reviewer #2: There are still spelling errors for example on page 38, acute should be accurate.

The time varying reproductive rate reference is not Pauline van den Dreissche's. Her article discusses ways to compute R0.

This article discusses time varying R0:

Massad E, Burattini MN, Lopez LF, Coutinho FA. Forecasting versus projection models in

epidemiology: the case of the SARS epidemics. Med Hypotheses. 2005;65(1):17-22. Epub

2005/05/17. doi: 10.1016/j.mehy.2004.09.029. PubMed PMID: 15893110; PubMed Central PMCID:

PMCPMC7116954.

7. PLOS authors have the option to publish the peer review history of their article (what does this mean?). If published, this will include your full peer review and any attached files.

Reviewer #2: **Yes: **Diana Thomas

---

## [Author Response · Author response to Decision Letter 1]

31 Jul 2020

Reviewer #2: 

1. There are still spelling errors for example on page 38, acute should be accurate.

Response: 

Thank you for the comment, the paper has been sent for editing. It is now OK. 

2. The time varying reproductive rate reference is not Pauline van den Dreissche's. Her article discusses ways to compute R0.

This article discusses time varying R0:

Massad E, Burattini MN, Lopez LF, Coutinho FA. Forecasting versus projection models in

epidemiology: the case of the SARS epidemics. Med Hypotheses. 2005;65(1):17-22. Epub

2005/05/17. doi: 10.1016/j.mehy.2004.09.029. PubMed PMID: 15893110; PubMed Central PMCID:

PMCPMC7116954.

Response: 

Thank you for the comment. We added the reference in the manuscript.

---

## [Editor Report · Decision Letter 2]

5 Aug 2020

Simulating the progression of the COVID-19 disease in Cameroon using SIR models

PONE-D-20-14463R2

Dear Dr. NGUEMDJO,

We’re pleased to inform you that your manuscript has been judged scientifically suitable for publication and will be formally accepted for publication once it meets all outstanding technical requirements.

Kind regards,

Yury E Khudyakov, PhD

Academic Editor

PLOS ONE
---

## [Editor Report · Acceptance letter]

7 Aug 2020

PONE-D-20-14463R2 

Simulating the progression of the COVID-19 disease in Cameroon using SIR models 

Dear Dr. Nguemdjo:

I'm pleased to inform you that your manuscript has been deemed suitable for publication in PLOS ONE. Congratulations! Your manuscript is now with our production department. 

Kind regards, 

on behalf of

Dr. Yury E Khudyakov 

Academic Editor

PLOS ONE